# Life and Death Decisions—The Many Faces of Autophagy in Cell Survival and Cell Death

**DOI:** 10.3390/biom12070866

**Published:** 2022-06-21

**Authors:** Ge Yu, Daniel J. Klionsky

**Affiliations:** 1Life Sciences Institute, University of Michigan, Ann Arbor, MI 48109-2216, USA; yuge@umich.edu; 2Department of Molecular, Cellular and Developmental Biology, University of Michigan, Ann Arbor, MI 48109-2216, USA

**Keywords:** cellular homeostasis, lysosome, mitophagy, neurodegeneration, stress

## Abstract

Autophagy is a process conserved from yeast to humans. Since the discovery of autophagy, its physiological role in cell survival and cell death has been intensively investigated. The inherent ability of the autophagy machinery to sequester, deliver, and degrade cytoplasmic components enables autophagy to participate in cell survival and cell death in multiple ways. The primary role of autophagy is to send cytoplasmic components to the vacuole or lysosomes for degradation. By fine-tuning autophagy, the cell regulates the removal and recycling of cytoplasmic components in response to various stress or signals. Recent research has shown the implications of the autophagy machinery in other pathways independent of lysosomal degradation, expanding the pro-survival role of autophagy. Autophagy also facilitates certain forms of regulated cell death. In addition, there is complex crosstalk between autophagy and regulated cell death pathways, with a number of genes shared between them, further suggesting a deeper connection between autophagy and cell death. Finally, the mitochondrion presents an example where the cell utilizes autophagy to strike a balance between cell survival and cell death. In this review, we consider the current knowledge on the physiological role of autophagy as well as its regulation and discuss the multiple functions of autophagy in cell survival and cell death.

## 1. Introduction to Autophagy

Living cells undergo a constant and dynamic turnover of their components; macromolecules, including proteins, lipids, and nucleic acids, and even entire organelles are being created and removed as they are damaged or as part of cellular remodeling in response to changing environmental conditions. In this way, the cell can degrade dysfunctional or superfluous parts of the cytoplasm before they accumulate and interfere with normal physiology.

There are two primary ways for the cell to eliminate cytoplasmic components: secretion or degradation. Compared to secretion, clearing constituents by degradation allows the cell to recycle the macromolecular breakdown products, such as amino acids. This is especially important when cells have limited access to nutrients or are under conditions of nutrient starvation. In addition, in the case of multicellular organisms, where the microenvironment plays an important role in maintaining cell survival in tissues, and where cell behavior is often regulated by secreted factors, such as hormones, clearing cellular constituents by secretion may not be optimal. Thus, removing cytoplasm by degradation can be an advantageous solution.

Autophagy is a well-conserved degradation process where cargoes are delivered to the vacuole in yeast or plants, or to lysosomes in more complex eukaryotic cells for degradation. Three primary types of autophagy have been identified in eukaryotic cells: macroautophagy, microautophagy, and chaperone-mediated autophagy [1]. The mechanisms of these three types of autophagy are different. In macroautophagy, cytoplasm is sequestered either specifically or non-specifically through the action of a transient compartment, termed a phagophore, which matures into a double-membrane vesicle called an autophagosome; this process is followed by the fusion of the autophagosome with the vacuole or lysosome, where the cargo is degraded. Microautophagy occurs without autophagosomes; cargoes are directly engulfed at the surface of the vacuole or lysosome by a stepwise invagination or protrusion and septation of the organelle membrane [1]. Both microautophagy and macroautophagy can sequester large protein complexes and organelles. In contrast, in chaperone-mediated autophagy, individual proteins containing a specific recognition motif are unfolded through the action of cytosolic chaperones and delivered into the lysosome lumen via translocation across the lysosome membrane with the help of lumenal chaperones and the receptor LAMP2A (lysosomal associated membrane protein 2A) [2]. For the remainder of the review, we focus on macroautophagy, and refer to it as autophagy.

The mechanism of autophagy is highly conserved from yeast to mammals (Figure 1). The most distinguishing morphological feature of autophagy involves dynamic membrane rearrangement during the step of sequestration and formation of the autophagosome; however, the entire process of autophagy, including the degradation and efflux steps, is typically required. Autophagy can be divided into the following steps [1,3]: (1) Induction. Autophagy occurs at a constitutive basal level and is upregulated in response to stress such as nutrient starvation. Under these conditions, the Atg1 complex/ULK1 complex (yeast/mammals) is activated by phosphorylation and, in turn, the Atg1/ULK1 kinase phosphorylates other components of the autophagy machinery, such as Atg13/ATG13 and Atg14/ATG14. (2) Vesicle nucleation. The Atg14-containing class III phosphatidylinositol (PtdIns) 3-kinase (PtdIns3K) complex I is recruited to the phagophore assembly site (PAS), which is proximal to the vacuole. Phosphatidylinositol-3-phosphate (PtdIns3P) is then produced at the PAS, allowing the recruitment of PtdIns3P-binding proteins, which in turn recruit additional downstream effector proteins. (3) Vesicle expansion and completion. These steps involve two ubiquitin-like conjugation systems that utilize Atg8/LC3/GABARAP conjugated to phosphatidylethanolamine (generating, for example, Atg8–PE/LC3-II) and the Atg12–Atg5-Atg16/ATG12–ATG5-ATG16L1 complex. Sealing of the double-membrane vesicle generates the mature autophagosome. (4) Fusion of the autophagosome with the vacuole/lysosome. This step involves tether and SNARE proteins similar to other vesicle-mediated fusion processes. (5) Degradation of cargoes and efflux of the breakdown products. The degradation step of autophagy is accomplished by resident acid hydrolases in the vacuole/lysosome. The wide repertoire of these hydrolases allows for the efficient breakdown and recycling of the cellular components delivered by autophagy; the resulting “building blocks”, such as amino acids, are released back into the cytosol through permeases for the cell to reuse in anabolic or catabolic pathways [4].

The packaging of cargoes in autophagy can be both nonselective and selective. The process of selective autophagy is very similar to nonselective autophagy, except that the autophagosome forms in close apposition around specific cargoes, excluding bulk cytoplasm. Such specificity is achieved by receptors that can bind or mark particular cargoes, targeting them for autophagic degradation. Autophagy receptors bind to a specific ligand, typically an integral part of the cargo, or the receptor itself may be an integral component of an organelle membrane. The receptor links the cargo to the autophagy machinery, in particular to Atg8–PE/LC3-II that is present on the concave surface of the phagophore; this process may also involve a scaffold protein [5]. In this way, phagophores can be recruited to specific cargoes. The type of cargo specificity provides the name for the selective autophagy process, such as mitophagy for mitochondria. The wide repertoire of targets in selective autophagy, including, but not limited to, organelles, pathogens, proteins, and lipid droplets, gives the process great potential in responding to various types of stress and signals to facilitate cell survival and prevent cell death [6]. The types of selective autophagy, the various receptors for specific cargoes, and the cargo recognition mechanism were systematically covered in a recent review [5].

Since the discovery of autophagy, its physiological role has been an intriguing question. Particularly, in terms of cell survival and cell death, how is autophagy beneficial? Given the recycling function of autophagy, as well as the well-established fact that autophagy is activated in response to various types of stress, such as nutrient starvation, the pro-survival role of autophagy has been long recognized. In recent years, it has been shown that the autophagy machinery is used for cellular activities that do not require vacuolar/lysosomal degradation, as described later in the review. These non-conventional functions of the autophagy machinery provide a deeper and more comprehensive understanding of the pro-survival role of autophagy (i.e., beyond supplying nutrients) and the genes involved. In addition to its pro-survival role, autophagy participates in and crosstalks with pathways in regulated cell death (RCD), providing an even wider application of the autophagy machinery. Decades of research have considerably pushed forward our understanding of the physiological role of this process. In the rest of this review, we consider the current knowledge regarding the role of autophagy in both cell survival and death. Importantly, in many cases discussed in this review, the ability of the autophagy machinery to participate in a variety of pro-survival and pro-death pathways depends on its ability to sequester, deliver, and degrade cargoes, emphasizing the importance of understanding the nature of the autophagy process.

## 2. Autophagy in Cell Survival

The early concept that autophagy can serve as a pro-survival pathway came from the observation that autophagy is regulated by nutrient conditions and is increased in response to a reduction in the glucose or amino acid level in cells or tissues [7,8]. As the research tools and model organisms used to study autophagy expanded, starvation-induced autophagy was observed across a wide range of eukaryotic organisms [9]. The importance of autophagy for survival can be most easily demonstrated with a single-cell organism such as yeast; autophagy-defective yeast mutants display a tremendous decrease in viability under starvation conditions [10]. Similarly, *atg5* knockout (KO) mice exhibit nutrient and energy insufficiency shortly after birth, accompanied by a significantly lower amino acid concentration in plasma and tissues despite their almost normal appearance upon birth. The early death of these mice can be rescued by milk ingestion [11].

Why is autophagy important for survival at the cellular level? Autophagy (or the autophagy machinery) facilitates cell survival in canonical and non-canonical ways. In canonical autophagy, two fundamental results are achieved: the degradation and recycling of cytoplasmic components. Although these two results are almost always coupled during the autophagy process, the cell could have a more urgent need for one versus the other under a given condition. For example, the cell can use selective autophagy to degrade unfavorable components, such as damaged organelles and pathogens. In these cases, the cell uses autophagy mainly for **clearance** (to prevent certain cellular components from causing further harm), but not for the degradation products gained from this process. In contrast, during nitrogen or amino acid starvation, the cell needs to acquire amino acids by degrading and recycling existing materials in the cell via autophagy. In these cases, it is the **recycling** function that is more important to the cell, and this is the main function of autophagy that most people envision when considering this process. In the following paragraphs, we discuss the applications of these two functions of autophagy. However, in addition to its canonical role in degradation, the autophagy machinery also facilitates cell survival in non-canonical ways that do not require the degradation step.

### 2.1. Autophagy for Clearance

Damaged macromolecules and organelles are generated over time and accumulate over the lifetime of the cell. If left unchecked, the accumulation of these components can lead to serious problems, such as DNA damage. By clearing these constituents, autophagy prevents them from causing further harm to the cell and thus facilitates cell survival. This clearance function of autophagy can be critical in terminally differentiated, non-dividing cells, such as neurons [12,13]. Both nonselective and selective autophagy promotes clearance of cytoplasmic components.

One example of autophagy in clearance is the removal of oxidized biomolecules (proteins, DNA, and lipids) when the cells are under oxidative stress [14]. Many signaling pathways, including those involving AMPK, MTOR, NFKB/NF-κB, HIF1A/HIF-1, and more, have been proposed to induce autophagy in response to oxidative stress [14,15,16,17]. Oxidative stress can also regulate autophagy more directly. Oxygen availability can affect the activity of KDM5C, which is responsible for the demethylation of ULK1. Upon hypoxia, the activity of KDM5C is reduced due to lack of oxygen, leading to the accumulation of symmetrical dimethylation at arginine 170 of ULK1. This modification of ULK1 activates its kinase activity, and thus increases autophagy flux [18].

Selective autophagy facilitates the turnover of damaged or superfluous organelles. For example, damaged mitochondria can lead to excessive reactive oxygen species, insufficient ATP supply, and/or apoptosis. In differentiated cells that cannot refresh their mitochondrial pool by cell division, mitophagy is proposed to be an important way to regulate mitochondria quality control and maintain homeostasis [12,13,19,20,21,22]. The accumulation of dysfunctional mitochondria has been hypothesized as one of the potential causes of Parkinson’s disease [23,24]. Autophagy can also participate in organelle homeostasis in response to environmental changes. For example, peroxisome biogenesis can be induced when yeast cells are incubated with oleic acid or methanol. If yeast cells are then provided with a preferred carbon source, such as glucose, cells use pexophagy (selective autophagy of peroxisomes) to degrade the excess peroxisomes [25,26].

In addition to its role in stress response, selective autophagy plays an integral part in development. For example, during yeast meiosis, the level of Atg40, the reticulophagy (selective autophagy of the endoplasmic reticulum [ER]) receptor, is induced, and reticulophagy is activated to remove a subset of this organelle. The loss of Atg40 and most of the Atg proteins leads to decreased sporulation efficiency [27]. In mammals, mitophagy is responsible for the clearance of mitochondria during the maturation of erythroid cells, and blocking mitophagy leads to reduced mature erythrocytes, causing anemia as well as shorter lifespan in mice [28,29]. The inherent roles of selective autophagy in development suggest a widespread role of this mode of autophagy.

Autophagy can also clear intracellular protein aggregates and pathogenic microbes. For example, ubiquitinated protein aggregates in yeast cells can be recognized by the receptor Cue5 via its ubiquitin-binding CUE domain, and its Atg8-family interacting motif/AIM recruits Atg8–PE. Recent research identified CCT2 as the autophagy receptor for polyQ-HTT (huntingtin) protein aggregates in mammalian cells [30]. Receptors, such as SQSTM1 and OPTN in mammalian cells, possess both a ubiquitin-binding domain and LC3-interacting region/LIR. These and other receptors are involved in the selective autophagic degradation of intracellular pathogens via xenophagy [31,32].

By fine-tuning selective autophagy, the cell can sequester/degrade different cargoes in response to different signals or stress. One of the many ways to achieve this is through the regulation of autophagy receptors. For example, the iron level in the cell can be regulated by ferritin, which can store and release iron. The level of NCOA4, the autophagy receptor for ferritinophagy (selective autophagy of ferritin), is regulated by ubiquitin-dependent proteasomal degradation in an iron-dependent manner. In response to iron depletion, NCOA4 is stabilized, facilitating the autophagic degradation of ferritin and hence iron release [33,34]. As will be described later in this review, mitochondria damage enhances the mitochondrial recruitment of mitophagy receptors, such as OPTN, to initiate mitophagy [13]. Importantly, mitophagy removes damaged mitochondria under mild stress and thus prevents intrinsic apoptosis mediated by the release of mitochondrial contents, such as CYCS/cytochrome c, into the cytosol, which is addressed later on in this review.

Although the sequestration and subsequent degradation of cargoes are almost always coupled in autophagy, temporal homeostasis can be restored solely by the sequestration step as it sequesters the unfavorable compartments from the cytoplasm, or the damaged parts from an organelle. This can be considered as an extension of the clearance role of autophagy. For example, it was reported that the clearance of dysfunctional parts of the ER via sequestration within an autophagosome may be sufficient to temporarily restore homeostasis without the subsequent degradation of the cargo [35]. Similarly, the sequestration of damaged parts of mitochondria into autophagosomes could prevent harm from excess reactive oxygen species without relying on the breakdown of the organelle.

### 2.2. Autophagy for Recycling

The recycling function of autophagy provides cells with raw material for synthesizing new molecules and producing energy. The degradation and recycling of cytoplasm are especially important when cells are experiencing stress, and are in nutrient-limiting conditions; however, it is critical to regulate autophagy to avoid excessive degradation that can lead to cell death. The regulation of autophagy under normal and stress conditions has been extensively studied, and there are excellent reviews that cover this topic [36,37,38,39]. Here, we briefly provide two examples of how the cell coordinates autophagy for its pro-survival role in response to stress, as well as the major regulatory pathways involved.

The target of rapamycin (TOR) pathway is one of the key regulators for autophagy in response to nutrient stress [37,40]. When nutrients are abundant, TOR complex 1 (TORC1/MTORC1) is active. In yeast, phosphorylation of Atg1 and Atg13 by active TORC1 inhibits the activity of the Atg1 complex. During nutrient starvation, TORC1 is inactivated, preventing the inhibitory phosphorylation of Atg1 and Atg13 and leading to increased autophagy induction. Similar to TORC1, mammalian MTORC1 can phosphorylate and inhibit proteins that participate in autophagy, including ULK1. TORC1/MTORC1 also regulates autophagy transcriptionally by directly or indirectly regulating the activity or localization of transcriptional regulators for *ATG* genes, underscoring the complexity of autophagy regulation in response to nutrient levels [37,39,40]. The regulation of TORC1/MTORC1 on autophagy also extends to other aspects, such as the mRNA stability of *ATG* genes and histone modification [37,39,40].

The AMP-activated protein kinase (AMPK) pathway promotes autophagy in response to energy stress. AMPK-mediated induction of autophagy facilitates the turnover of macromolecules to provide substrates for catabolic pathways [41]. AMPK is activated when the cellular AMP:ATP ratio is high [42,43,44]. Activated AMPK phosphorylates and activates many autophagy-related proteins, including ULK1, BECN1 (a homolog of yeast Vps30/Atg6), ATG9A, and subunits of the class III PtdIns3K complex. AMPK also inhibits the formation of other different PIK3C3/VPS34-containing complexes involved in autophagy-independent processes [45]. In addition, AMPK stimulates autophagy by mediating the upregulation of autophagy-related genes through the phosphorylation of transcription factors. For example, AMPK phosphorylates FOXO3 (forkhead box O3), promoting its nuclear translocation and transcriptional activity; FOXO3 target genes include *ULK1*, *ATG4B*, *ATG14*, *ATG12*, *BECN1*, and *BNIP3*. AMPK can also suppress MTOR activity, which, as described above, can lead to an increase in autophagy [39,41,42].

## 3. Pro-Survival Roles of the Autophagy Machinery Independent of Clearance or Vacuolar/Lysosomal Degradation

In addition to its primary role in degradation, the autophagy machinery performs multiple tasks. From a general viewpoint, the autophagy machinery can (1) **recognize** specific cargoes, (2) **partition** the cargo(es) into a double-membrane structure, and (3) **deliver** the double-membrane structure to the vacuole/lysosome. In addition to degradation, this process can also be viewed as a general partitioning and delivery system in the cell. In the following paragraphs, we describe the role of the autophagy machinery in the cytoplasm-to-vacuole targeting (Cvt) pathway and secretory autophagy. These pathways expand the pro-survival function of the autophagy machinery.

### 3.1. The Cvt Pathway

Most resident vacuolar proteins are delivered to this organelle through a portion of the traditional secretory pathway, from the ER to the Golgi apparatus and then being diverted to endosomes and the vacuole. Yeast cells can also utilize autophagy to deliver proteins in the cytosol to the vacuole through the Cvt pathway. The Cvt pathway is biosynthetic, taking enzymes from their site of synthesis in the cytosol to their site of function, the vacuole, but it essentially uses the autophagy machinery for transport [5,46]. During this process, specific vacuolar enzymes are recognized by autophagy receptors, followed by the recruitment of core autophagy machinery for sequestration within a double-membrane Cvt vesicle; as with other types of selective autophagy, the Cvt vesicle is closely apposed to the cargo (e.g., precursor aminopeptidase I [prApe1]) and excludes bulk cytoplasm. The completed Cvt vesicle fuses with the vacuole to deliver the cargo, which in this case is not degraded, but may be subject to proteolytic activation. The cargo specificity in the Cvt pathway is accomplished by specific cargo receptors. For example, Atg19 is the receptor for the hydrolases Ams1, prApe1, and Ape4. An autophagic mechanism thus serves as a delivery system in the Cvt pathway, ensuring proper vacuole function [5]. The discovery of the Cvt pathway is an important complement to the understanding of the function of the autophagy machinery. While this pathway is not found in mammalian cells, the secretory autophagy pathway, as described below, is similar to the Cvt pathway, as both pathways selectively sequester specific cargoes from the cytoplasm and deliver them to a specific destination.

### 3.2. Secretory Autophagy

Secretory autophagy utilizes both the partitioning and delivery functions of autophagy. While many secreted proteins rely on the signal sequence at their N terminus to enter the ER and follow the conventional ER–Golgi secretory pathway, for secreted proteins that lack a signal sequence, unconventional secretion pathways are used [47,48]. One such pathway is secretory autophagy, in which cytoplasmic secretory cargoes are engulfed within autophagosomes, which then fuse with the plasma membrane instead of lysosomes. One of the advantages of secretory autophagy is that it can secrete proteins directly from the cytoplasm [49,50].

One of the well-established cargoes identified in secretory autophagy is IL1B (interleukin 1 beta). IL1B is subject to sequestration in LC3-positive vesicles, followed by the fusion of such vesicles with the plasma membrane [51]. IL1B is recruited to phagophores via its interaction with TRIM16. TRIM16 interacts with the R-SNARE SEC22B for its recruitment to LC3-II^+^ sequestration membranes [52]. SEC22B on the secretory autophagosome interacts with SNARE proteins SNAP23, SNAP29, and STX3 (syntaxin 3) to allow fusion with the plasma membrane to accomplish the secretion of IL1B [52]. SEC22B may have a critical role in directing the secretory autophagosome away from the degradative pathway, causing it to fuse with the plasma membrane instead of a lysosome [50,52].

Caution should be taken when using the term “secretory autophagy”. Sometimes this term is confused with the term “autophagy-dependent secretion” [50]; however, these two processes are not necessarily identical. For example, the secretion of some cargoes depends on autophagy, but does involve autophagosomes; rather, autophagy degrades molecules that suppress their secretion. This distinction is especially important when interpreting genetic interactions between autophagy and the secretion of a cargo. Despite this caveat, it is important to acknowledge that the secretion of some aggregation-prone proteins is affected by autophagy, including amyloid beta (Aβ), and SNCA/α-synuclein, further emphasizing the role of autophagy in cell survival [49,50].

In addition to its relatively well-characterized role in the Cvt pathway and secretory autophagy, the autophagy machinery (or autophagy genes) has been implicated in many other cellular events, ranging from DNA repair to cell division to T-cell differentiation. A recent review provides a more comprehensive description of the non-canonical functions of the autophagy machinery [53], which are becoming a new research hotspot [54,55,56].

## 4. Autophagy in Cell Death

Autophagy is associated with cell death originally because of electron microscopy-based observations of autophagic structures in dying cells [57,58]. Some of these reports have been controversial as the observation of autophagy proximate to cell death could be due to a pro-survival activation of autophagy (antagonizing cell death) instead of a pro-death role [59,60,61]. Continued research has confirmed the participation of autophagy in cell death progression in *Dictyostelium discoideum*, *Caenorhabditis elegans*, *Drosophila*, and mammalian cell lines [60]. For example, mutation in many autophagy genes in *C. elegans*, including *atg3*, *atg2*, or *atg5*, reduces germ cell death induced by gamma radiation [62]. Similarly, autophagy is required for cell death induced by treatment with chemotherapeutic drugs in *Bax*- and *Bak1*-deficient mouse embryonic fibroblasts [63]. Although the specifics of the experimental manipulation may limit the interpretation in regard to the physiological role of autophagy in cell death, this research not only indicates the capability of the autophagy process in participating in cell death, but also provides insight into the role of autophagy under pathologically relevant conditions.

### Autophagy-Dependent Cell Death

Regulated cell death (RCD) refers to cell death caused by genetically encoded mechanisms for the elimination of cells that are irreversibly damaged, superfluous, and/or potentially harmful, or for the programmed elimination of cells during development. RCD operates at the level of an organism or a colony to eliminate useless or (potentially) harmful cells, or allow dying cells to release molecules that alert the organism or colony about a threat [61]. Recent research has shown that autophagy plays an essential or facilitating role in multiple types of RCD. According to the Nomenclature Committee on Cell Death, autophagy-dependent cell death (ACD) is defined as a type of RCD that relies on the autophagic machinery or components thereof. In addition to autophagy-dependent cell death, autophagy also facilitates the process of ferroptosis, FAS-driven extrinsic RCD, necroptosis, and autosis (a specific instance of autophagy-dependent cell death) [61].

Many findings of ACD are obtained from studies on development, as these studies circumvent the caveats of human intervention, such as using RCD-inducing drugs. The regulated cell death of *Drosophila* salivary glands and midgut cells during development are well-established examples of ACD. The RCD of *Drosophila* salivary glands is reduced when either apoptosis or autophagy is blocked. The removal of salivary glands is further delayed when both RCD and autophagy are delayed, indicating that both processes are functional in this event [64]. The developmental RCD of *Drosophila* midgut can be severely delayed by mutation or knockdown of *Atg1*, *Atg2*, or *Atg18*, but not by inhibition of apoptosis [65]. Consistent with autophagy playing a role during salivary or midgut RCD, the expression of many *Atg* genes is upregulated during these two processes [65,66]. An increased number of GFP-Atg8a puncta (corresponding to autophagosomes, Atg8a is the functional Atg8 paralog in *Drosophila melanogaster*) under confocal microscopy, as well as an increased number of autolysosome-like structures under transmission electron microscopy, are observed during developmental RCD of *Drosophila* midgut [67].

Genetic studies using RNAi-mediated screening indicate that there is a deviation in the use of enzymes in ACD of *Drosophila* midgut from the canonical autophagy process. Specifically, Atg3, involved in lipidation of Atg8a, and Atg7, involved in both the Atg8 and Atg12 conjugation steps, are not required for RCD in *Drosophila* midgut [68]. Instead, another E1 enzyme in ubiquitination, Uba1, is required in this process [67]. Along these lines, Atg5, Atg12, and Atg16, which act together as an E3 enzyme for Atg8a conjugation, are also not required during this process; nor are Atg6 and Atg14, both of which are involved in the nucleation step of autophagy, despite the fact that these proteins are all required for starvation-induced autophagy in *Drosophila* fat body cells [60]. Whether or not the context-specific usage of proteins in autophagy is part of the tight regulation of this process requires further investigation.

The detailed mechanism by which autophagy facilitates ACD remains to be illustrated. Because autophagy flux relies on lysosomal degradation, it is likely that autophagy-dependent cell death also requires the lysosome [60,69,70]. The involvement of autophagy in development indicates that the cell has a flexible utilization of molecules involved in this process, but many questions remain regarding the involvement of autophagy in cell death. For example, does autophagy function to degrade major cellular compartments to achieve cell death? Does autophagy promote cell death by degrading survival factors? Is selective autophagy or nonselective autophagy required for ACD?

## 5. Crosstalk between Autophagy and RCD

The dual role of autophagy in cell survival and cell death makes the question of the interplay between autophagy and RCD more intriguing and important. Recent research has revealed many examples of crosstalk between these two pathways. In mouse hepatocytes that are depleted of *Casp1*, there is a decrease in the autophagy markers LC3 and BECN1, as well as decreased clearance of mitochondria compared to wild-type cells; overexpressing BECN1 restores the clearance of mitochondria [71]. TP53/p53, a major tumor suppressor that can induce cell death, interacts with RB1CC1/FIP200 (a potential human homolog of Atg17) and BECN1 [72,73]. These results indicate there is crosstalk between autophagy and RCD.

Further investigation uncovered some of the details of the crosstalk between these two pathways. A number of autophagy proteins are substrates of caspases. For example, stimulation of death receptors, such as FAS and TNFRSF10A/TRAILR1, by death ligands, such as TNF/TNF-α or TNFSF10/TRAIL, leads to a significant increase in ATG3 protein degradation by the initiator CASP8, leading to inhibition of autophagy during extrinsic apoptosis [74]. Similarly, initiator caspases CASP9 and CASP10, as well as executioner caspases CASP3 and CASP6, can cleave ATG5, leading to decreased autophagy flux [75].

Whereas some of the known autophagy-related caspase substrates experience a loss of function in autophagy upon cleavage by caspases, human ATG4D and BECN1 are proposed to gain pro-apoptotic functions after being cleaved. Among the four human paralogs ATG4A to ATG4D, ATG4D contains a canonical caspase cleavage sequence (DEVD63K) in its N terminus [76]. ATG4D is cleaved by CASP3 in vitro [76]. Researchers also observed an increased ANXA5/annexin V staining in HeLa cells overexpressing ΔN63 ATG4D, suggesting a role of this ATG4D fragment in apoptosis. Consistent with ATG4D being a caspase substrate in vitro, in human A431 cells and HeLa cells treated with the apoptosis inducer staurosporine, full-length ATG4D is lost, and this can be prevented by co-treatment with the caspase inhibitor Z-VAD-FMK. Interestingly, mutating the catalytic site does not abolish the effect of ΔN63 ATG4D on apoptosis. Further investigation showed that ATG4D can be imported to the mitochondrial matrix, regardless of caspase cleavage. Upon treatment with the mitochondrial uncoupler carbonyl cyanide m-chlorophenyl hydrazone (CCCP), the mitochondrial pool of ATG4D sensitizes cells to cell death [77]. Together, these data suggest a role of ATG4D in apoptosis independent of its already known function in autophagy.

BECN1 also demonstrates an interesting balance between its roles in autophagy and apoptosis: (1) BECN1 interacts with BCL2-family proteins via the BH3 domain, positioning itself at the interface between autophagy and apoptosis [78]. The association of BECN1 with the BCL2 family inhibits the pro-autophagic role of BECN1 [79]. (2) BECN1 is the substrate of multiple caspases, including CASP3, and CASP6 to CASP10 [75], and caspase cleavage inhibits BECN1-mediated autophagy. Interestingly, the C-terminal cleavage product of BECN1 can localize at the mitochondria and sensitize the cells to apoptosis, suggesting that caspase cleavage of BECN1 helps coordinate autophagy and apoptosis by inhibiting the former and promoting the latter [79,80].

Conversely, regulators of RCD can also be controlled by autophagy or autophagy gene products. Depletion of the *Atg5* gene significantly represses apoptosis in mouse embryonic fibroblasts treated with SKI-I (a pan-sphingosine kinase inhibitor), accompanied by suppressed CASP8 activity. Upon SKI-I treatment, CASP8 is recruited to ATG5-positive autophagic membranes, leading to the suggestion that autophagy facilitates the activation of this caspase [81]. ATG7 interacts with CASP9 in multiple human tumor cell lines. Overexpressing ATG7 in living cells inhibits the processing of the CASP9 prodomain, suggesting that ATG7-CASP9 interaction sequesters CASP9 and prevents its activation [82]; whereas, in a cell-free system, an increased dose of ATG7 inhibits CASP9 protease activity [82]. In cells undergoing apoptosis, knockdown of ATG12 inhibits BAX activation and CYCS/cytochrome-c release from mitochondria. Additionally, the antiapoptotic effect of the MCL1 protein can be mitigated by the expression of ATG12, but not the expression of ATG12^V95A^, a mutant defective in binding to MCL1 [83].

What is the biological relevance of having crosstalk between autophagy and RCD? One of the advantages could be that such crosstalk helps the cell make a definitive “survival” and “death” decision under a given condition and avoid wasting energy. Compared to executing both the pro-survival autophagy and pro-death RCD in response to any stress, it would be more beneficial for the cell to execute only autophagy under mild stress allowing cell survival, and switch to mainly RCD under strong stress conditions, or in response to developmental needs. As mentioned above, the binding of ATG7 to CASP9 weakens the activity of the latter. However, in response to the apoptosis-inducing drug staurosporine, the interaction of ATG7-CASP9 is disrupted, freeing CASP9 from inhibition [82]. In this way, cells coordinate the two pathways not only to avoid the waste of energy in having two antagonistic pathways functioning at the same time, but also to focus on one pathway under a given condition. From this perspective of cost and efficiency, perhaps it is not surprising to see that many molecules regulate both autophagy and RCD in a coordinated manner. Another potential advantage of having such a steep gradient of response concerns the context of tissue damage. A steeper switch in the survival or death decision can help limit the damaged area by executing RCD where needed, while allowing cells with less damage to use autophagy to survive. Such a model is proposed for ischemic infarction [78].

The explanation above, however, does not explain findings, such as the pro-apoptotic role of ATG12 via binding to BCL2 [83]. Thus, the interplay between autophagy and cell death appears to be more complex and intricate. During the course of scientific research, many of the genes involved in the autophagy and RCD pathways were originally named based on the pathways they were first identified within. The pro-death role of autophagy indicates a more complex picture and inspires researchers to view the autophagy process and the genes involved from different perspectives, and the same applies to genes involved in regulated cell death. Consistent with this notion, there are many genes “moonlighting” in both pathways (Figure 2).

### The Mitochondrion Is One of the Juncture Points for Autophagy and RCD

As the target of mitophagy, and a platform to initiate apoptosis, the mitochondrion is another convergent point between autophagy and RCD. Mitochondria contain apoptotic factors, such as ENDOG (endonuclease G) and CYCS/cytochrome c. During intrinsic apoptosis, mitochondrial outer-membrane permeabilization (MOMP) is changed, leading to changes in mitochondrial membrane potential and the release of apoptotic factors from mitochondria [61]. For example, once released into the cytosol, CYCS/cytochrome c can interact with and cause the oligomerization of APAF1, forming the apoptosome to initiate apoptosis. The MOMP is regulated by effector BCL2-family proteins. In response to an intrinsic apoptosis stimulus, the pro-apoptotic effector BCL2-family proteins form oligomers in the mitochondrial outer membrane, causing the release of certain factors, such as CYCS/cytochrome c [61].

Changes in mitochondrial membrane potential can also trigger mitophagy in mammalian cells. Mitophagy can be induced by treating cells with either CCCP, which reduces mitochondrial membrane potential, or a combination of antimycin A and oligomycin, which inhibit the electron transport chain and ATP synthase, respectively [84]. These drugs can lead to the depolarization of mitochondrial membrane potential. This depolarization inhibits the cleavage of PINK1 and its subsequent degradation, leading to its stabilization on the mitochondrial outer membrane. The stabilized PINK1 recruits PRKN/Parkin, an E3 ubiquitin ligase, which then ubiquitinates mitochondrial proteins. The formation of ubiquitin chains on mitochondrial proteins results in the binding of autophagy receptors, such as OPTN (optineurin), CALCOCO2/NDP52, and the RAB signaling proteins RABGEF1, RAB5, and RAB7A, to the mitochondrial surface, marking the organelle for mitophagy [13,85].

As mitophagy degrades mitochondrial contents via lysosomal degradation, it could be used to antagonize mitochondria-mediated intrinsic apoptosis, which requires the release of mitochondrial contents, such as CYCS/cytochrome c, into the cytosol. The clearance of damaged mitochondria by mitophagy can therefore increase the threshold for apoptosis initiation. The antagonizing effect of mitophagy and intrinsic apoptosis allows the cells to degrade just one or a few mitochondria when the stress is mild to avoid the execution of apoptosis. For example, the inhibition of mitophagy promotes B5G1-induced apoptosis in drug-resistant cancer cells [86]. However, this notion might be too simple to reflect the full relationship between mitophagy and apoptosis, as PINK1 can regulate BCL2-family proteins, and PRKN activation can both facilitate or inhibit cell death [87,88]. As mitochondria perform multiple functions from energy and metabolite production to regulating apoptosis, it is not surprising that mitochondrial turnover is under complicated regulation and that there is an intricate crosstalk between the pathways involved. The crosstalk between mitophagy and apoptosis has been intensively studied, and the relationship between the two pathways has also been covered by previous reviews [88].

## 6. Conclusions

Autophagy has both pro-survival and pro-death functions for the cell, endowed by its natural abilities to recognize, sequester cytoplasmic contents into double-membrane structures, and deliver them to a subcellular compartment having the opposite topology (the lumen of the lysosome/vacuole being equivalent to the extracellular environment). As a pro-survival pathway, autophagy (or the autophagy machinery) facilitates cell survival in both canonical and noncanonical ways. As a degradation pathway, canonical autophagy facilitates cell survival via (1) clearing unfavorable components in the cell, such as damaged organelles, oxidized biomolecules, and protein aggregates, preventing them from causing further harm; (2) recycling existing components to supply the cell with building blocks, such as amino acids, which can be used for new molecular synthesis and energy production. Degradative autophagy is subject to complex regulation, allowing the cell to flexibly adjust the targeted cargoes and the flux of autophagy according to different kinds of stress and developmental signals. The inherent partitioning and delivery functions of the autophagy machinery suggest the potential means of utilization. Indeed, recent research has shed light on the degradation-independent roles of the autophagy machinery, such as its essential role in the biosynthetic Cvt pathway in yeast and in secretory autophagy in mammalian cells. As a pro-death pathway, it has been shown that autophagy participates in multiple forms of RCD, including ACD, ferroptosis, FAS-driven extrinsic RCD, necroptosis, and autosis. The context-dependent differences in autophagy protein usage in ACD from canonical autophagy have been indicated, and an intricate crosstalk between autophagy and RCD has been shown. Such crosstalk, as well as the role of autophagy in both cell survival and cell death, is exemplified in the number of genes shared by the two pathways, and in the complex balance between mitophagy and apoptosis, emphasizing the importance of the coordination between the two pathways, which could be beneficial for the cell to reduce energy cost and make a definitive “survival” or “death” choice. Together, this research suggests a wider implication of the autophagy machinery and gene products involved in many cellular events, and the many faces of autophagy in cell survival and cell death. Originating from the studies on its primary role in degradation, we are now able to better appreciate the nature of the autophagy machinery beyond its long-recognized role in degradation, and the various roles of autophagy this nature endows. Along with ongoing investigation into genes involved in the autophagy process, the studies on how this process coordinates with other pathways to achieve a fine-tuned autophagic behavior as well as flexible utilization of the autophagy machinery will be a major focus in the future in order to appreciate the functions of autophagy in cell survival and cell death.

## Figures and Tables

**Figure 1 biomolecules-12-00866-f001:**
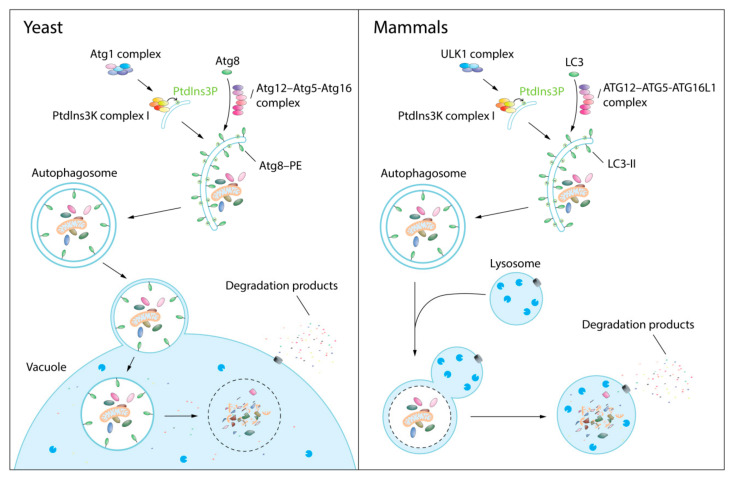
The autophagy process in yeast and mammalian cells. When autophagy is induced, the activated Atg1/ULK1 (yeast/mammals) complex activates downstream effector proteins, including Atg14/ATG14. The activated Atg14/ATG14-containing class III phosphatidylinositol (PtdIns) 3-kinase (PtdIns3K) complex I then produces phosphatidylinositol-3-phosphate (PtdIns3P) at the phagophore assembly site to promote vesicle nucleation. With the help of the Atg12–Atg5-Atg16/ATG12–ATG5-ATG16L1 complex, Atg8/LC3 is conjugated to phosphatidylethanolamine (PE) at the phagophore membrane to generate Atg8–PE/LC3-II. The phagophore expands and engulfs cytoplasmic cargoes, forming a mature autophagosome. The autophagosome is then fused with the vacuole in yeast cells or a lysosome in mammalian cells, where the cargoes are degraded by acid hydrolases. The degradation products, such as amino acids, are released back to the cytosol by permeases in the vacuolar or lysosomal membrane.

**Figure 2 biomolecules-12-00866-f002:**
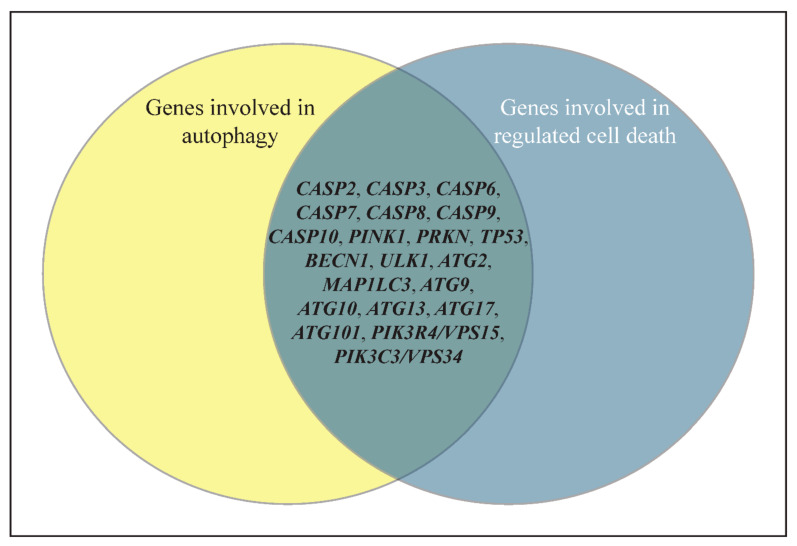
A Venn diagram of some genes that are shared between the autophagy and regulated cell death (RCD) pathways based on the published literature. Although the names of the genes usually reflect the pathways they were originally identified within (autophagy or regulated cell death), it is the functions of the genes/proteins that determine how and to what extent a gene can be utilized by the cells. The details of the role of these genes in autophagy and RCD can be found in the text and the references in this review [60,75].

## Data Availability

Data sharing not applicable.

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
