# Peer review of "Life and Death Decisions—The Many Faces of Autophagy in Cell Survival and Cell Death"

_biomolecules, 2022, doi:10.3390/biom12070866_

Round 1

Reviewer 1 Report

This is a comprehensive, well-written, and extremely clear reviw; ann added value of the text is that it is not a mere a “telephone list” of biological and medical scenarios where autophagy and cell death play a role or overlaps, but it critically analyses the interconnections between these two phenomena. The cited literature is updated and covers the most important papers in the field. I have just a couple of minor observations:

-       An interesting point raised by the authors is the different perspectives researchers looked at genes and pathways, and the influence of this different perspective had on nomenclature, as highlighted in figure 2. In my opinion, this point could be better addressed if in Figure 2 the shared genes and proteins would colored in different ways depending on where the area of research they come from, or by adding genes  non shared in the non-overlapping areas of the Venn diagram

-       In some cases. I suggest the authors back up their observations, not by other  reviews -although extremely well written and prestigious – but by the original research articles (see for examples ref. 39-42)

Author Response

This is a comprehensive, well-written, and extremely clear reviw; ann added value of the text is that it is not a mere a “telephone list” of biological and medical scenarios where autophagy and cell death play a role or overlaps, but it critically analyses the interconnections between these two phenomena. The cited literature is updated and covers the most important papers in the field. I have just a couple of minor observations:

  1. “An interesting point raised by the authors is the different perspectives researchers looked at genes and pathways, and the influence of this different perspective had on nomenclature, as highlighted in figure 2. In my opinion, this point could be better addressed if in Figure 2 the shared genes and proteins would colored in different ways depending on where the area of research they come from, or by adding genes  non shared in the non-overlapping areas of the Venn diagram.”

           We think this is an interesting suggestion; however, when we tried either of these modifications we were not satisfied with the results for various reasons. For example, one concern is that autophagy-related genes that have not yet been implicated in cell death may be shown in the future to have such a role. By placing them outside of the overlapping area we would be implying that they definitely have no role in cell death. A second problem is that BECN1 was originally identified based on its role in apoptosis; however, most people now think of it as an autophagy-related gene. Coloring BECN1 in some way to indicate that it came from apoptosis would probably be misleading or seem confusing to most people. Thus, we think it is best to essentially leave this figure as it remains (although we have made a few minor corrections). At any rate, we have added a sentence to the figure legend to address the reviewer’s comment.

  1. “In some cases. I suggest the authors back up their observations, not by other  reviews -although extremely well written and prestigious – but by the original research articles (see for examples ref. 39-42).”

We agree with the reviewer and have added additional citations of the primary research throughout the article.

Reviewer 2 Report

In this review, Ge Yu and Daniel J. Klionsky summarize  the current understanding  on autophagy physiology and regulation; they also highlight the role of autophagy in cell survival and cell death.

The work summarizes with a precise and elegant language the varied physiological functions of autophagy as well as its interconnection with the mechanisms of cell survival and cell death.

I strongly recommend its publication.

Author Response

In this review, Ge Yu and Daniel J. Klionsky summarize  the current understanding  on autophagy physiology and regulation; they also highlight the role of autophagy in cell survival and cell death.

The work summarizes with a precise and elegant language the varied physiological functions of autophagy as well as its interconnection with the mechanisms of cell survival and cell death.

I strongly recommend its publication.

            We appreciate the positive comments of the reviewer.